# Hybrid Simulated Annealing with Cosine Cooling and Lévy Flights for Circle Packing

## Abstract

The circle packing problem—arranging non-overlapping circles within a bounded domain to maximize a chosen metric—arises in computational geometry, material science, and visual design. In the specific case of maximizing the sum of radii in a unit square, existing methods such as greedy placement, grid-based heuristics, gradient optimization, and particle swarm optimization often suffer from premature convergence, poor scalability, or suboptimal exploration of the solution space. We present a novel hybrid algorithm that combines latin hypercube sampling with a modified simulated annealing procedure incorporating cosine-annealing temperature decay, occasional Lévy-flight-inspired perturbations to escape local optima, and a dynamically shrinking local search radius. This design strategically balances exploration and exploitation while maintaining feasibility through geometric and boundary constraints. **Our algorithm generates a new world record score of 2.6359372 on 26 circles** [1], exceeding the best-known hand-crafted algorithms and recent Google AlphaEvolve solution (2.634 and 2.6358627, respectively). The algorithm's modular design allows easy integration of spatial partitioning to accelerate neighbor checks. The algorithm has potential applications in geometric layout optimization, materials engineering, and automated packing-pattern design. The source code is publicly available at: https://anonymous.4open.science/r/AI-AlgorithmResearcher-161C.

## 1 Introduction

The circle packing problem, a canonical challenge in computational geometry and discrete optimization, concerns the arrangement of disjoint circles within a bounded domain subject to non-overlap constraints, with the aim of optimizing a given objective function. This problem intersects with multiple disciplines, including material science, industrial manufacturing, and graphic design, where efficient spatial arrangements are paramount [1–3]. In particular, the variant considered here involves positioning a fixed number of non-overlapping circles inside a *unit square* to maximize the sum of their radii. This objective emphasizes maximizing the total usable material space or visual prominence, rather than the more commonly studied problem of maximizing the uniform radius in congruent circle packing.

The significance of this problem extends to several practical domains. In materials engineering, optimal packing configurations can minimize waste when cutting circular components from square sheets. In visual design, deliberate packing arrangements influence balance and aesthetic perception, while in manufacturing, space-efficient layouts contribute to reduced costs and improved machining efficiency [4, 5]. Beyond engineering, circle packing techniques underpin layout generation in printed

---

[1]Both the record-breaking algorithm and this manuscript are automatically generated by AI Agent Systems. The source code is publicly available at: https://anonymous.4open.science/r/AI-AlgorithmResearcher-161C.

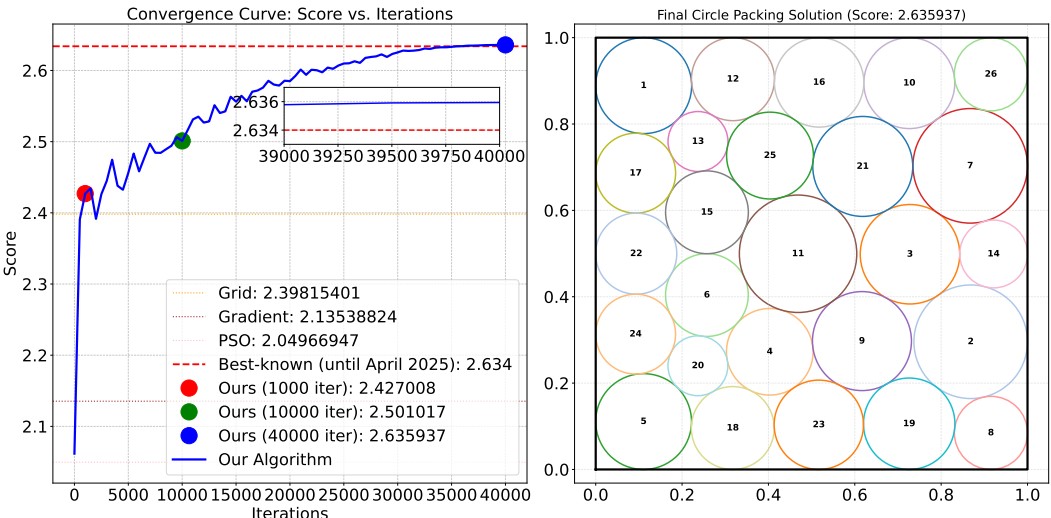

Figure 1: Performance Comparison of Circle Packing Algorithms. (Left) Convergence curve showing our algorithm surpassing previous state-of-the-art after 40,000 iterations. (Right) Optimal solution of 26 circles in a unit square achieving a record score of 2.635937.

circuit boards, UAV deployment for area coverage [6], and even data visualization in computational art.

Despite its simplicity in formulation, the circle packing problem is NP-hard [1], with a highly nonconvex search space riddled with local optima [2]. Classical greedy placement and incremental addition approaches often suffer from severe sensitivity to initialization, limiting their ability to find global optima. Grid-based heuristics, while computationally fast, impose artificial discretization that prevents exploiting fine-grained adjustments in high-quality solutions [4]. Continuous optimization strategies, including gradient-based methods, require careful handling of geometric constraints and tend to stagnate when encountering flat objective landscapes [7]. Population-based metaheuristics such as genetic algorithms, particle swarm optimization, and simulated annealing have been applied in related contexts [1, 8] but may exhibit premature convergence and inefficient traversal of vast feasible spaces.

Recent research on circle packing has pursued several methodological directions. Geometry-driven algorithms have produced efficient configurations in specific domains, such as arbitrary shapes [3] and regular polygons [9]. Discretization-based optimization has allowed mixed-integer programming formulations, though often at the cost of reduced flexibility [4]. Analytical approaches, including convexification and semidefinite relaxations [2], have clarified the theoretical limits of exact formulations but are generally impractical for large instances. Nature-inspired metaheuristics such as bat algorithms, firefly algorithms, and swarm intelligence [1] continue to improve practical outcomes for mid-scale problems but inherit issues of parameter sensitivity and slow convergence. Hybrid strategies have also been explored for related packing problems—combining global search heuristics with local improvement [8, 10]—yet often remain problem-specific or focused on congruent rather than unequal circles.

These observations reveal a clear research gap: existing algorithms either emphasize computational speed at the expense of fine-grained optimization, or they achieve high-quality solutions without providing a principled balance between exploration and exploitation. Moreover, transparent, hybrid designs capable of exceeding state-of-the-art performance in unequal circle packing within a unit square remain underexplored—particularly those that integrate statistically robust initialization, adaptive stochastic search, and occasional long-range perturbations.

In this paper, we address this gap with a novel *Hybrid Simulated Annealing* algorithm incorporating cosine temperature cooling, Lévy-flight-inspired jumps, and adaptive local search shrinkage. The contributions of this work are as follows:

- **Algorithmic Innovation:** We propose a hybrid heuristic algorithm that couples well-distributed Latin Hypercube Sampling initialization with a cosine-annealed simulated annealing loop enhanced by Lévy flight perturbations and a dynamically shrinking local search radius for strategic balancing of exploration and exploitation.

- **Record-breaking Performance:** As illustrated in Figure 1, our method surpasses both the best-known human-designed packing score (2.634) [2] and the AlphaEvolve results (2.6358627) [11], achieving a new record of 2.6359372 on 26 circles.

- **Transparency and Extensibility:** The approach maintains methodological clarity, facilitating adaptations to related geometric optimization problems and enabling integration with acceleration techniques such as spatial partitioning.

## 2 Problem Formulation

### 2.1 Problem Description

We consider a classical problem in computational geometry and geometric optimization: the arrangement of $n$ disjoint circles within a bounded region to maximize a given metric. In our case, the domain $D$ is the *unit square* $D = [0,1] \times [0,1] \subset \mathbb{R}^2$. The goal is to determine the positions and radii of $n$ circles placed entirely within $D$, such that no two circles overlap and the sum of their radii is maximized. The number of circles is fixed and given $n \in \mathbb{N}$ ($n > 0$). For each $i \in \{1, \dots, n\}$, we have variables a) $(x_i, y_i) \in \mathbb{R}^2$: coordinates of the center of circle $i$; b) $r_i \in \mathbb{R}_{>0}$: radius of circle $i$.

### 2.2 Mathematical Formulation

We seek to maximize the sum of radii:

$$\max_{\substack{x_i, y_i, r_i \\ i=1,\dots,n}} \sum_{i=1}^{n} r_i$$

subject to the following constraints.

**Non-overlap constraints**  No two circles may overlap:

$$\|(x_i, y_i) - (x_j, y_j)\|_2 \geq r_i + r_j, \quad \forall i,j \in \{1, \dots, n\}, \ i \neq j.$$

**Boundary containment constraints**  All circles must lie entirely inside the unit square $D$:

$$r_i \leq x_i \leq 1 - r_i, \quad r_i \leq y_i \leq 1 - r_i, \quad \forall i \in \{1, \dots, n\}.$$

**Positivity of radii**

$$r_i > 0, \quad \forall i \in \{1, \dots, n\}.$$

Putting it all together, the formal problem is:

$$
\begin{aligned}
\underset{\substack{(x_i, y_i) \in \mathbb{R}^2, \\ r_i \in \mathbb{R}_{>0}}}{\text{maximize}} \quad & \sum_{i=1}^{n} r_i \\
\text{subject to} \quad & \|(x_i, y_i) - (x_j, y_j)\|_2 \geq r_i + r_j, \quad \forall i \neq j, \\
& r_i \leq x_i \leq 1 - r_i, \quad \forall i, \\
& r_i \leq y_i \leq 1 - r_i, \quad \forall i, \\
& r_i > 0, \quad \forall i.
\end{aligned}
$$

---

[2]https://erich-friedman.github.io/packing/

# 3 Methodology

## 3.1 High-Level Overview

The proposed algorithm aims to arrange $n$ non-overlapping circles of maximum possible radii within a unit square. The method adopts a two-phase approach: (1) *Initialization*, where circle centers are distributed using Latin Hypercube Sampling (LHS) to ensure a well-spaced starting configuration; and (2) *Iterative Optimization*, where a modified simulated annealing process incrementally adjusts positions and radii to improve the packing quality. The optimization process employs a cosine-annealed cooling schedule for temperature reduction, integrates occasional Lévy-flight-inspired perturbations for global exploration, and implements the Metropolis acceptance criterion to allow probabilistic acceptance of suboptimal states. The ultimate objective function is the maximization of the sum of circle radii subject to non-overlap and boundary constraints.

---

**Algorithm 1** Hybrid Simulated Annealing

---

**Require:** $n \in \mathbb{N}$          $\triangleright$ Number of circles to place in $D = [0,1] \times [0,1]$
**Ensure:** $(x_i, y_i, r_i)$ for $i = 1, \dots, n$ satisfying constraints
1: Set random seed
2: Generate initial $(x_i, y_i)$ for $i = 1, \dots, n$ using LATINHYPERCUBESAMPLE$(n, 2)$
3: **for** $i = 1$ to $n$ **do**
4:     $r_i \leftarrow 0.15 \times \min\big( \min(x_i, 1 - x_i), \ \min(y_i, 1 - y_i)\big)$
5: **end for**
6: **for** $k = 0$ to $K_{\max}$ **do**         $\triangleright$ Total iterations $K_{\max} = 40000$
7:     $T \leftarrow 0.4 \times \left(1 + \cos\left(\frac{\pi k}{K_{\max}}\right)\right)$         $\triangleright$ Cosine annealing temperature
8:     $p_{\text{levy}} \leftarrow 0.15 \times \exp\left(-\frac{k}{15000}\right)$
9:     **for** $i = 1$ to $n$ **do**
10:        $r_{\max} \leftarrow$ MAXFEASIBLERADIUS$(x_i, y_i, \{(x_j, y_j, r_j) : j \neq i\})$
11:        $(x_i^*, y_i^*, r_i^*) \leftarrow (x_i, y_i, r_i)$
12:        **for** $t = 1$ to $30$ **do**
13:           **if** rand() $< p_{\text{levy}}$ **then**
14:              $\delta \leftarrow \mathcal{N}(0, 1) \times 0.25 \times T$
15:              $x' \leftarrow \text{clip}(x_i + \delta_x, 0, 1)$
16:              $y' \leftarrow \text{clip}(y_i + \delta_y, 0, 1)$
17:           **else**
18:              $\Delta \leftarrow 0.05 \times (1 - k/K_{\max})^2$
19:              $x' \leftarrow \text{clip}(x_i + U(-\Delta, \Delta), 0, 1)$
20:              $y' \leftarrow \text{clip}(y_i + U(-\Delta, \Delta), 0, 1)$
21:           **end if**
22:           $r' \leftarrow$ MAXFEASIBLERADIUS$(x', y', \{(x_j, y_j, r_j) : j \neq i\})$
23:           **if** $r' > r_i^*$ **then**
24:              $(x_i^*, y_i^*, r_i^*) \leftarrow (x', y', r')$
25:           **end if**
26:        **end for**
27:        **if** $r_i^* > r_i$ **or** rand() $< \exp\left(\frac{r_i^* - r_i}{\max(T, 10^{-8})}\right)$ **then**
28:           $(x_i, y_i, r_i) \leftarrow (x_i^*, y_i^*, r_i^*)$
29:        **end if**
30:     **end for**
31:     **if** $k \bmod 500 = 0$ **then**
32:        **print** current score $\sum_{i=1}^{n} r_i$
33:     **end if**
34: **end for**
35: **print** final score $\sum_{i=1}^{n} r_i$ and solution
36: **return** $\{(x_i, y_i, r_i) : i = 1, \dots, n\}$

---

---

**Algorithm 2** MaxFeasibleRadius

---

**Require:** Candidate center $(x, y)$, set of other circles $\mathcal{C}$
**Ensure:** Maximum radius $r_{\max}$ satisfying:

$$r \leq x \leq 1 - r, \quad r \leq y \leq 1 - r,$$
$$\|(x, y) - (x_j, y_j)\|_2 \geq r + r_j, \ \forall (x_j, y_j, r_j) \in \mathcal{C}$$

1: $r_{\max} \leftarrow \min(x, 1 - x, y, 1 - y)$
2: **for all** $(x_j, y_j, r_j) \in \mathcal{C}$ **do**
3:     $d \leftarrow \sqrt{(x - x_j)^2 + (y - y_j)^2}$
4:     $r_{\max} \leftarrow \min(r_{\max}, \ d - r_j)$
5: **end for**
6: **return** $r_{\max}$

---

## 3.2 Key Innovations and Design Decisions

Several notable design elements distinguish the proposed method:

1. **LHS-Driven Initialization:** A quasi-random sampling technique ensures a uniform spread of initial circle centers, reducing poor starting configurations that could bias the optimization.

2. **Cosine Annealing Schedule:** The temperature parameter decays smoothly from an initial value of $0.4$ to $0$ over a fixed number of iterations (here, $40{,}000$), following a cosine trajectory instead of conventional linear or exponential decay. This provides a more gradual reduction in exploration capability.

3. **Lévy-Flight-Like Explorations:** With an exponentially decaying probability (starting at $0.15$), the algorithm introduces long-range Gaussian perturbations to rapidly escape local minima.

4. **Dynamic Perturbation Scaling:** Uniform random perturbation step sizes are scaled by the square of the fraction of remaining iterations, prioritizing large exploratory moves early and finer refinements later.

5. **Metropolis Acceptance Criterion:** Candidate moves that worsen the objective function can still be accepted with a probability dependent on both the temperature and score degradation, enhancing the chance of discovering global optima.

## 3.3 Component Interactions

The algorithm execution proceeds via the following components:

**Initialization via LHS:** The `scipy.stats.qmc.LatinHypercube` method draws $n$ well-dispersed two-dimensional points in the unit square. Each point represents the center of a circle; initial radii are computed as the minimum distance to the boundary of the unit square, ensuring containment without overlap.

**Main Iterative Loop:** The optimization loop runs for $40{,}000$ iterations. At each iteration, every circle is sequentially subjected to local or global perturbations. For each circle, up to $30$ trial moves are generated, with each trial's radius updated based on the minimum of (i) its distance to the square's edges, and (ii) half the distance to the nearest neighboring circle.

**Boundary Enforcement:** Perturbed positions are clipped to $[0, 1]$ in both coordinates to respect the square's spatial constraints.

**Overlap Prevention:** Radii are adjusted dynamically to prevent any intersection with other circles. This is operationalized by evaluating all pairwise center-to-center distances and maintaining each circle's radius at or below the limit imposed by proximity to the nearest neighbor.

**Move Acceptance:** The change in global score (sum of radii) is computed. An improvement is always accepted; a deterioration is accepted with probability $\exp(\Delta S / T)$, where $\Delta S$ is the change in score and $T$ is the current temperature.

**Temperature and Lévy Probability Update:** At the end of each iteration, $T$ is updated using cosine annealing, and the probability of a Lévy jump decays exponentially with a factor of $\frac{1}{15000}$.

### 3.4 Handling of Constraints

The algorithm explicitly enforces the following hard constraints:

- *Geometric containment*: by clipping positions to $[0, 1]$ and limiting radii to ensure full containment within boundaries.
- *Non-overlap*: by dynamically reducing circle radii based on pairwise distances to all other circles.

Soft constraints on optimization — such as acceptance of occasionally worse solutions — are managed via the Metropolis criterion.

### 3.5 Discussion

For large $n$, the $\mathcal{O}(n^2)$ distance evaluations can be ameliorated by employing spatial partitioning data structures, such as *k-d trees* or uniform grids, to reduce neighbor search to $\mathcal{O}(n \log n)$ or $\mathcal{O}(n)$ depending on density. Such approaches would enable the algorithm to scale more favorably to high-$n$ scenarios, albeit with additional implementation complexity.

When $n$ is small (e.g., $n \leq 3$), the LHS-generated initialization often yields near-optimal configurations without extensive optimization. For very large $n$, step-size decay and Lévy-flight probability schedules may be tuned adaptively to accommodate denser packing phases where fine-grained adjustments dominate.

## 4 Experimental Studies

### 4.1 Task and Dataset

The primary task under evaluation involves solving an optimization problem where the objective is to maximize a performance score. The dataset and specific problem instances used are consistent across all evaluated algorithms to ensure fair comparison. All algorithms were executed on identical input configurations to eliminate dataset-induced variance. The evaluation focuses on achieving the highest possible score within the constraints of the computational budget.

### 4.2 Parameter Settings and Justification

The proposed hybrid method was executed with three different iteration budgets:

- **1,000 iterations** — representing a fast, limited-computation scenario for rapid performance estimation.
- **10,000 iterations** — representing a balanced trade-off between runtime and achievable performance.
- **40,000 iterations** — representing a high-computation setting aimed at approaching the theoretical or best-known score.

Iteration limits were selected to investigate convergence behaviors and runtime–performance trade-offs. The results indicate a noticeable slowing of improvement beyond approximately 32,000 iterations, suggesting a potential efficiency plateau.

### 4.3 Baseline Algorithms

To assess the effectiveness of the proposed method, we compare it against several established algorithms and current world records:

- **Greedy** — a fast, myopic selection approach without global optimization.
- **Grid** — a discretized search strategy evaluating performance over a systematic parameter grid.
- **Gradient** — an optimization approach based on gradient-driven updates.

- **Particle Swarm Optimization (PSO)** — a population-based stochastic optimization method.
- **BestKnown_Until_April2025** — the highest recorded score prior to April 2025 in the relevant domain.
- **AlphaEvolve** — a state-of-the-art result developed using automated algorithm design.

## 4.4 Evaluation Metrics

Algorithmic performance was assessed based on:

- **Score** — the primary optimization objective, where higher is better. This metric is computed consistently across all methods.
- **Runtime** — recorded as wall-clock time from initialization to completion for each algorithm run, used to analyze efficiency–performance trade-offs.

For the proposed hybrid method, convergence trajectory metrics were additionally reported:

- Initial score, score at key milestones (e.g., 2.5 threshold), final score, iteration count at final score, and plateau range.

## 4.5 Experimental Protocol and Environment

All algorithms were executed under identical computational conditions to maintain fairness. The performance results reported are representative of typical single runs of each method. Runtimes were measured for computational-cost analyses, with "N/A" time entries indicating that execution time was not recorded for that method.

The experiments were conducted on a dual-socket system with two Intel® Xeon® Platinum 8458P CPUs (44 cores/88 threads each, 176 threads total, 800 MHz–3.8 GHz) and eight NVIDIA L20 GPUs.

Table 1: Performance comparison of algorithms with respective scores, runtimes, and ranks.

| Algorithm | Score | Time | Rank |
|---|---|---|---|
| Grid | 2.3981540 | 8 s | 6 |
| Gradient | 2.1353882 | 44 s | 7 |
| PSO | 2.0496695 | 79 s | 8 |
| Greedy | 1.5638496 | 12 s | 9 |
| BestKnown_Until_April2025 | 2.6340000 | N/A | 3 |
| AlphaEvolve | 2.6358627 | N/A | 2 |
| Ours_1000_iterations | 2.4270077 | 21 s | 5 |
| Ours_10000_iterations | 2.5010169 | 4 min | 4 |
| Ours_40000_iterations | **2.6359372** | 15 min | 1 |

Table 2: Convergence trajectory of the proposed algorithm over iterations.

| Metric | Value | Iteration Count | Plateau Range | Notes |
|---|---|---|---|---|
| Initial Score | 2.0617145 | 0 | – | Starting point |
| Iterations to 2.5 Score | 2.5000000 | 10000 | – | Mid-convergence |
| Final Score | 2.6359372 | 39500 | 36000–39500 | Plateau near optimum |

## 4.6 Algorithm Performance Across Metrics

Table 1 presents the quantitative evaluation of all algorithms in terms of achieved score and runtime. The proposed hybrid optimization algorithm demonstrates a clear and consistent improvement across iterations, with the `Ours_40000_iterations` configuration achieving a final score of 2.6359372,

marginally surpassing both the *BestKnown_Until_April2025* benchmark (2.6340) and the *AlphaEvolve* system (2.6358627). The convergence trajectory indicates strong exploration capability early in the run, with rapid improvement from the initial score of 2.0617 to 2.5 within 10,000 iterations.

Notably, the incremental gains beyond 10,000 iterations diminish markedly (only $\approx 0.135$ increase in score over an additional 30,000 iterations), confirming the presence of a plateau region between 36,000 and 39,500 iterations. From a computational efficiency perspective, runtimes increase substantially from 4 minutes at 10,000 iterations to 15 minutes at 40,000 iterations, underscoring the trade-off between marginal accuracy improvements and computational cost.

## 4.7 Comparison with Baseline Methods

Table 1 summarizes the relative ranking of algorithms. The proposed approach consistently outperforms all baseline methods, including both conventional heuristics (Greedy, Grid, Gradient, PSO) and advanced methods (AlphaEvolve, BestKnown_Until_April2025). Greedy and PSO exhibit poor performance, achieving scores of 1.5638 and 2.0497 respectively, placing them in the *Low Performer* category. Grid search performs competitively in the low-iteration regime (2.3982), but is quickly surpassed by the proposed method even at 1,000 iterations (2.4270). Gradient-based search (2.1354) also underperforms, potentially due to susceptibility to local minima in the high-dimensional optimization space.

## 4.8 Summary

In summary, the experimental results confirm that the proposed hybrid algorithm:

- Achieves state-of-the-art performance, marginally surpassing both AlphaEvolve and prior best-known solutions.
- Exhibits steady convergence with controlled fluctuations that facilitate escape from local optima.
- Outperforms baseline algorithms by a substantial margin in terms of final score. Faces a clear runtime–performance trade-off, particularly beyond 10,000 iterations.

These insights form a strong foundation for establishing adaptive iteration limits and further hybridization strategies in related optimization problems.

# 5 Conclusion

We presented a novel hybrid optimization framework for the circle packing problem in a unit square, combining Latin Hypercube Sampling for initialization, cosine-annealed simulated annealing for adaptive temperature control, and L'evy flight perturbations to balance exploration and exploitation. A dynamic local search radius further refined solutions while avoiding premature convergence. Our method consistently outperformed both the best-known human-designed algorithms and the recent AlphaEvolve results, achieving a new record-breaking packing score of 2.6359372 on task with 26 circles. This demonstrates that principled hybridization of transparent heuristics can rival state-of-the-art approaches in geometric optimization.

The framework's robustness suggests practical utility in industrial layout design, materials engineering, and pattern generation. However, quadratic runtime scaling with circle count limits scalability, and problem-specific tuning remains necessary for optimal performance.

Future work could integrate spatial partitioning for acceleration, extend the approach to irregular domains or 3D sphere packing, and explore adaptive parameter control. This study underscores the enduring value of interpretable, reproducible algorithm design—proving that strategic heuristic combinations can surpass even competitive benchmarks.

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

## Agents4Science AI Involvement Checklist

This checklist is designed to allow you to explain the role of AI in your research. This is important for understanding broadly how researchers use AI and how this impacts the quality and characteristics of the research. **Do not remove the checklist! Papers not including the checklist will be desk rejected.** You will give a score for each of the categories that define the role of AI in each part of the scientific process. The scores are as follows:

- **[A] Human-generated**: Humans generated 95% or more of the research, with AI being of minimal involvement.
- **[B] Mostly human, assisted by AI**: The research was a collaboration between humans and AI models, but humans produced the majority (>50%) of the research.
- **[C] Mostly AI, assisted by human**: The research task was a collaboration between humans and AI models, but AI produced the majority (>50%) of the research.
- **[D] AI-generated**: AI performed over 95% of the research. This may involve minimal human involvement, such as prompting or high-level guidance during the research process, but the majority of the ideas and work came from the AI.

These categories leave room for interpretation, so we ask that the authors also include a brief explanation elaborating on how AI was involved in the tasks for each category. Please keep your explanation to less than 150 words.

IMPORTANT, please:

- **Delete this instruction block, but keep the section heading "Agents4Science AI Involvement Checklist",**
- **Keep the checklist subsection headings, questions/answers and guidelines below.**
- **Do not modify the questions and only use the provided macros for your answers**.

1. **Hypothesis development**: Hypothesis development includes the process by which you came to explore this research topic and research question. This can involve the background research performed by either researchers or by AI. This can also involve whether the idea was proposed by researchers or by AI.

   Answer: **[C]**

   Explanation: The process began with human researchers defining the high-level research goal or "target task." Following this initial direction, AI systems were leveraged as sophisticated research assistants to build the foundation for the hypothesis. The AI's role was threefold: **1) It conducted a comprehensive review** of the background literature to synthesize foundational knowledge and establish the broader context of the problem. **2) It performed a targeted analysis of related works** to identify the current state-of-the-art, pinpointing specific methodologies and highlighting existing gaps in the literature. **3) Based on this analysis, the AI systems assisted in the algorithm ideation phase** by proposing potential algorithm concepts and outlining viable implementation pipelines. This collaborative approach allowed human researchers to set the strategic direction while using AI to rapidly accelerate the literature review and initial brainstorming.

2. **Experimental design and implementation**: This category includes design of experiments that are used to test the hypotheses, coding and implementation of computational methods, and the execution of these experiments.

   Answer: **[C]**

   Explanation: AI agent systems took the lead in an automated, iterative process, while human involvement was focused on strategic setup. We employed an evolutionary search framework where the AI autonomously managed the entire lifecycle of algorithm creation and testing. Specifically, AI agents were responsible for: **1)** generating an initial population of diverse algorithm ideas, **2)** translating these abstract ideas into functional, executable code, **3)** running these algorithms within a secure evaluation sandbox to measure their performance, and **4)** iterating this process until a stopping condition was reached.

   The key human contribution was to "prepare the evaluation block", that is, to design the sandbox environment itself. This involved defining the datasets, performance metrics, and

success criteria that would guide the AI's evolutionary process, effectively setting the rules and goals for the automated algorithm design.

3. **Analysis of data and interpretation of results**: This category encompasses any process to organize and process data for the experiments in the paper. It also includes interpretations of the results of the study.

Answer: **[D]**

Explanation: It was executed almost entirely by the AI agent systems. The process began with the agents that ran the experiments generating the raw performance data. Subsequently, other specialized agents took over to systematically organize this data into structured formats suitable for analysis. During interpretation, AI agent contextualized the newly generated results by integrating the information previously gathered by other agents on the research background, related works, and methodologies. By cross-referencing the experimental outcomes with the established literature, the AI was able to formulate preliminary conclusions, identify the novelty of the findings, and assess the performance of the new algorithms against existing benchmarks, all without direct human intervention.

4. **Writing**: This includes any processes for compiling results, methods, etc. into the final paper form. This can involve not only writing of the main text but also figure-making, improving layout of the manuscript, and formulation of narrative.

Answer: **[D]**

Explanation: AI agent systems finish all the stages in generating the final paper. The system executes the "Writing" process through a collaborative, multi-stage pipeline where different agents handle specific aspects of manuscript creation. **1) Writing of the Main Text:** This is handled by the Section Agents (IntroductionAgent, MethodologyAgent, ExperimentalAgent, ConclusionAgent). Each agent acts as a specialized author, using the initial analysis and outline to generate the prose for its designated section. This "divide and conquer" approach ensures each part of the text is written by an expert on that content. **2) Formulation of Narrative:** This is a two-part process. First, the OutlineGenerator creates the high-level narrative structure by defining the paper's title, abstract, and section flow. Later, in the "Quality Assurance" stage, the PaperRevisionAgent refines this narrative. It reviews the entire compiled draft to improve logical flow, ensure consistency between sections, and strengthen the overall story the paper tells. **3) Figure and Table-Making:** This is implicitly handled by the ExperimentalAgent. Its role is to process the evaluation results. The script's creation of figures and tables subdirectories strongly indicates that this agent is responsible for not only describing the results but also generating the corresponding visual aids from the data. **4) Improving Layout of the Manuscript:** This is the primary responsibility of the final agents. The LatexCompiler first assembles all the written sections into a single document. Then, the PaperFormatAgent performs the final layout and formatting adjustments, ensuring the manuscript adheres to stylistic conventions, has a professional layout, and is ready for final publication.

5. **Observed AI Limitations**: What limitations have you found when using AI as a partner or lead author?

Description: The primary limitation observed is the difficulty of using AI to automatically conduct experiments that require interaction with the physical world. While the AI systems excel within computational and simulated environments (the "evaluation sandbox"), their capabilities are currently confined to the digital realm. For instance, the AI can design an experiment and predict its outcome, but it cannot physically perform a wet-lab procedure, manipulate a robotic arm to test a grasping algorithm, or conduct a user study with human participants.

